# Decision-Making Preferences among Advanced Cancer Patients in a Palliative Setting in Jordan

**DOI:** 10.3390/ijerph20085550

**Published:** 2023-04-18

**Authors:** Omar Shamieh, Ghadeer Alarjeh, Mohammad Al Qadire, Waleed Alrjoub, Mahmoud Abu-Nasser, Fadi Abu Farsakh, Abdelrahman AlHawamdeh, Mohammad Al-Omari, Zaid Amin, Omar Ayaad, Amal Al-Tabba, David Hui, Eduardo Bruera, Sriram Yennurajalingam

**Affiliations:** 1Department of Palliative Care, King Hussein Cancer Center, Amman 11941, Jordan; 2Center for Palliative & Cancer Care in Conflict, King Hussein Cancer Center, Amman 11941, Jordan; 3Faculty of Medicine, The University of Jordan, Amman 11941, Jordan; 4Faculty of Nursing, Al al-Bayt University, Mafraq 25113, Jordan; 5College of Nursing, Sultan Qaboos University, Muscat 123, Oman; 6Department of Medicine, King Hussein Cancer Center, Amman 11941, Jordan; 7Office of Nursing, King Hussein Cancer Center, Amman 11941, Jordan; 8MD Anderson Cancer Center, Houston, TX 77030, USA

**Keywords:** Jordan, decision control, palliative care, cancer, patient satisfaction, communication

## Abstract

Understanding patients’ decision-making preferences is crucial for enhancing patients’ outcomes. The current study aims to identify Jordanian advanced cancer patients’ preferred decision-making and to explore the associated variables of the passive decision-making preference. We used a cross-sectional survey design. Patients with advanced cancer referred to the palliative care clinic at a tertiary cancer center were recruited. We measured patients’ decision-making preferences using the Control Preference Scale. Patients’ satisfaction with decision-making was assessed with the Satisfaction with Decision Scale. Cohen’s kappa statistic was used to assess the agreement between decision-control preferences and actual decision-making, and the bivariate analysis with 95% CI and the univariate and multivariate logistic regression were used to examine the association and predictors of the demographical and clinical characteristics of the participants and the participants’ decision-control preferences, respectively. A total of 200 patients completed the survey. The patients’ median age was 49.8 years, and 115 (57.5%) were female. Of them, 81 (40.5%) preferred passive decision control, and 70 (35%) and 49 (24.5%) preferred shared and active decision control, respectively. Less educated participants, females, and Muslim patients were found to have a statistically significant association with passive decision-control preferences. Univariate logistic regression analysis showed that, being a male (*p* = 0.003), highly educated (*p* = 0.018), and a Christian (*p* = 0.006) were statistically significant correlates of active decision-control preferences. Meanwhile, the multivariate logistic regression analysis showed that being a male or a Christian were the only statistically significant predictors of active participants’ decision-control preferences. Around 168 (84%) of participants were satisfied with the way decisions were made, 164 (82%) of patients were satisfied with the actual decisions made, and 143 (71.5%) were satisfied with the shared information. The agreement level between decision-making preferences and actual decision practices was significant (ⱪ coefficient = 0.69; 95% CI = 0.59 to 0.79). The study’s results demonstrate that a passive decision-control preference was prominent among patients with advanced cancer in Jordan. Further studies are needed to evaluate decision-control preference for additional variables, such as patients’ psychosocial and spiritual factors, communication, and information sharing preferences, throughout the cancer trajectory so as to inform policies and improve practice.

## 1. Introduction

Patients with advanced cancer usually face several events requiring them to either make decisions or participate in decision-making [1], especially when cancer progresses to advanced stages, patients are faced with treatment failure, or when there is a deterioration in the patient’s condition. Hence, contributions from patients or family caregivers to the decision-making process are frequently sought. Patient-centered shared decisions can improve the quality of cancer care, improve patients’ satisfaction with care, and reduce the cost of care [2,3]. Patients’ involvement in their medical care is needed. So, exploring their preferences regarding participation in the decision-making process is crucial.

Decision-making preference is “an individual’s expectation of having the power to participate in decisions to obtain desirable consequences” [4]. Three models of decision-making are recognized in clinical practice: passive decision-making, where patients delegate decisions to their family and/or physician; active decision-making, as patients make decisions regarding their medical care; and the shared decision-making approach, which involves patients, their families, and their primary physician [5].

In Western culture, individual autonomy is valued, and active decision-making is the most common preferred approach. However, the passive decision-making approach is found to be prevalent in several countries, including, but not limited to, India and Jordan [5,6,7,8,9,10]. Patients’ preference for decision-making is frequently associated with cultural background, beliefs, the healthcare system, patient expectations, satisfaction, and paternalistic approaches in medical decision-making, in addition to patients’ socio-demographic factors such as age, gender, education, and employment status [5,6,7,8,9,10].

A previously published international cohort cross-sectional survey involving 11 countries (USA, Jordan, Philippines, France, Singapore, South Africa, India, Brazil, Chile, Argentina, and Spain) with a total of 1490 participants revealed that the participants preferred an active role in decision-making. Additionally, there was a significant variation in the frequency of passive decision-control preferences by patient’s education, performance status, and country of origin [9]. Patients with a better performance status; higher education; and from Brazil, South Africa, and Jordan were significantly associated with the preference for passive decision-making [9]. These variations in decision preferences need more exploration in order to understand the factors that influence these preferences.

A French study enrolled 200 patients with cancer and showed that 37.7% preferred passive decision control, followed by active (36.2%) and then shared decision control (26.1%) [11]. In this study, patients with higher education levels, employed, and who were younger preferred active or shared decision-making approaches [11].

In another pooled analysis of four studies to assess the preferred decision-making approach among 7169 German patients with prostate cancer, the majority of patients (62.2%) preferred shared decision-making. In addition, younger patients with high quality of life scores opted for an active decision-making approach [12].

Exploring cancer patients’ preference regarding decision-making forms the first step toward patient-centered care. In some Eastern cultures, paternalistic medical approaches are commonly used as a form of decision-making, for example a comparative research design study conducted in Jordan recruited 86 oncology doctors concluded that although doctors in this study valued the importance of patients’ self-determination, autonomy, and patients’ right to have adequate health-related information, they still had their paternalistic approach in decision-making, where they tended to underuse the shared decision-making approach with patients. In this study, older doctors with more experience were found to be more comfortable using the shared decision approach than younger and less experience doctors [13].

In Middle Eastern countries, including Jordan, decision-control research is scarce for cancer patients in general and advanced cancer patients in particular [2,13,14]. Understanding cancer patients’ decision preferences is crucial to enhance their satisfaction and for individualized quality of care [9]. It may also lead to better communication and less conflicts among patients and healthcare professionals. In Jordan, only one study has explored the preference for decision-making among women with breast cancer [2]. Furthermore, three studies have explored preference regarding the disclosure of cancer diagnosis and information needed about their disease [6,7,13]. More studies within the Eastern Arabic Islamic culture that involve a heterogeneous group of patients with cancer are needed. Thus, the current study’s primary purpose is to examine the decision-making preferences of Jordanian patients with advanced cancer.

The main aim of this study was to identify the preferred and actual decision-making practices of patients with advanced cancer in Jordan, identify the associated variables of passive decision-making among this group of participants, and assess their satisfaction regarding information sharing and the decision-making process concerning their healthcare.

## 2. Materials and Methods

### 2.1. Study Design and Participants

A cross-sectional survey design was used. This study was part of an international multicenter study, in which Jordan was among 11 countries [9]. We used the convenience sampling approach to recruit patients with advanced cancer who were referred to the palliative care clinic at a tertiary cancer center. We included patients who were 18 years or older, diagnosed with advanced cancer (metastatic, locally advanced, or recurrent cancers), had at least one consultation encounter with the palliative care team and had no cognitive impairment as assessed by the clinicians using the Diagnostic and Statistical Manual of Mental Disorders (4th ed.; DSM-IV), were able to understand and speak Arabic or English, and were willing to participate in the study.

### 2.2. Study Setting

Our study was conducted at a tertiary cancer center in Amman, Jordan. The center provides cancer care to Jordanians and non-Jordanians from neighboring countries such as Syria, Iraq, Palestine, Yemen, Libya, and Sudan [15]. The center treats more than 60% of cancer cases in Jordan, and provides all cancer care modalities, including surgery, chemotherapy, radiotherapy, targeted therapy, and immunotherapy, in addition to bone marrow transplantation for adults and pediatrics. The center contains the region’s largest oncology palliative care program, offering all models of care, including an acute palliative care unit, ambulatory palliative care clinics, in-patient palliative care consultation service, hospice, and palliative home care services. The care is delivered through an interdisciplinary team and follows international standards and guidelines adapted to patient needs [16].

Jordan can be considered as a role model for other low- and middle-income countries, in that, despite limited resources, Jordan delivers high quality, patient-centered services, possesses world class health care professionals, and has an excellent regional and international reputation including cancer care [15]. In 2018, there were 9248 patients diagnosed with cancer. Cancer care is fully covered by governmental public insurance for all Jordanians [17]. In Jordan, cancer care is delivered through public hospitals, including military services, Ministry of Health, University hospitals, King Hussein Cancer Center, and the private sector [15].

### 2.3. Four Questionnaires Were Used to Collect Data in this Study

A demographic data sheet that was designed to collect patients’ demographical characteristics, including age, gender, level of education, employment, religion, marital status, and patients’ clinical data regarding cancer diagnosis and the received treatment modality (radiation, chemotherapy, immunotherapy, surgery, and targeted treatment).Control Preferences Scale (CPS): Participation preferences were measured by CPS, developed and validated by Denger and colleagues [18,19]. The tool was used in many studies to assess the decision-control preferences of patients with cancer; it is composed of four questions. The first question is about the preferred decision control for patient care by asking patients how the decisions regarding their care should be made. Patients were asked to choose only one option from the 15 given answers that were listed below the question. Based on the answers, the patients had passive, active, or shared decision control. Patients who selected any options from 1 to 4 opted for an active role, 5–12 opted for passive, and 13–15 opted for shared decision control (Appendix A) [9,18,19]. The second question assessed the actual decision control by asking patients how the decisions about their care were taken. Patients were allowed to choose only one answer from the 15 options listed. Based on patients’ answers, the actual decision-making process was later described as shared, passive, or active decision control. Patients who answered 1–4 were active, answers from 5−12 were passive (from 5–8 family made the decision and from 9–12 it was made by the doctor); answers from 13–15 indicated shared decision control. If patients chose number 13, the decision was shared between the doctor and patient; for answer 14, the decision control was shared between patient and family; and for answer 15, the decision control was shared between the family, patient, and doctor (Appendix A). The third and fourth questions were used to examine patients’ preferences regarding the physician’s or family’s involvement in their decision-making process. Each question had five options from 1–5. The patients were asked to choose one option only. Accordingly, the patients who chose option 1 or 2 were categorized as active, option 3 as shared, and option 4 or 5 as passive decision-control preferences (Appendix A) [9,18,19].Satisfaction with Decision Scale (SWDS): SWDS was used to assess the degree of patients’ satisfaction with the information they received about their care, how decisions about their care were made, and with the decisions themselves. It was originally developed by Holmes-Rovner, Margaret, et al., 1996. It contains six Likert-type scale items. It is a reliable (Cronbach’s alpha = 0.86) and valid scale. Its content, criterion, and discriminative validity were established [20]. Patients were asked to rate their response from 0, strongly disagree, to 4, strongly agree. Patients were unsatisfied if they chose options 0 or 1, satisfied if they chose 3 or 4, and undecided if they chose 2 (Appendix A).Karnofsky Performance Scale (KPS): KPS is a valid and reliable tool developed in 1949 by Dr. Bruchenal and Dr. Karnofsky. It is used broadly by healthcare providers and in many studies to assess cancer patients’ performance status by covering 11 stages from 100% (normal health) to 0% (death), decreasing by 10 points in each stage. Patients with 80–100% performance meant that they had a normal performance, they could do their daily activity with no need for help from others; 50–70% meant that they needed help in daily activity; while ≤40% meant that they needed continuous assistance and might deteriorate to reach death more rapidly (Appendix A) [21].

### 2.4. Instruments Translation

CPS and SWDS were translated to Arabic by two bilingual research team members. Then, they were back-translated to English by another two independent local translators to determine the linguistic and semantic equivalence by comparing the Arabic and the English versions. Furthermore, the approved Arabic version was subjected to an expert panel for final approval. Finally, the agreed-on version was used in this study.

### 2.5. Data Collection Procedure

The research team screened patients visiting palliative outpatient clinics for eligibility criteria. All eligible participants were approached in person to explain the study’s purpose and requirements. Then, if they agreed to participate in the study, they were asked to sign the consent form and fill the required questionnaires with the help of the research team in a designated clinic room in order to ensure patient comfort and privacy. Before embarking on the study, it was approved by the King Hussein Cancer Center (KHCC) institutional review board (proposal no. 13 KHCC 62).

### 2.6. Data Analysis

Data were entered and analyzed using Statistical Analysis Software SAS version 9.4 (SAS Institute Inc., Cary, NC, USA). Descriptive statistics such as mean, standard deviation, median, and interquartile ranges (IQR) were used for numerical variables. Frequencies (i.e., numbers) and percentages were calculated for the categorical variables. Cohen’s kappa statistic was used to assess the agreement between the decision-control preferences and actual decision-making. Bivariate analysis with 95% CI, using chi-square tests, were used to examine the association between the demographical and clinical characteristics of the participants (i.e., age, gender, employment, level of education, treatment modality, and KPS) and the participants’ decision-control preferences among Jordanian patients with advanced cancer. Finally, univariate and multivariate logistic regression were used to predict the relationship between the statistically significant demographical and clinical characteristics and participants’ decision-control preferences, using active versus passive and shared preferred decision control as the outcome variable in the model. For the primary objective of the original international multi-center study, the MD Anderson research team estimated the proportion of passive decision-control preference and the 95% CI for each country. A *p*-value < 0.05 was considered statistically significant. Each country should have a minimum of 100 patients allowing a 95% CI ± 9%. For example, a country for which they calculated a proportion of 30% passive decision would have a 95% CI of (21%, 39%). A total of 1490 patients were enrolled from all sites. In KHCC, the Jordan site, we planned to enroll ≥200 participants to allow for site-specific analysis.

## 3. Results

### 3.1. Sample Characteristics

The research team collected data from 2014 to 2015; 299 patients met the eligibility criteria and were approached. Of them, 99 patients refused to participate for being fatigued (*n* = 45), lack of interest in the study (*n* = 20), or having no time to complete the study questionnaire (*n* = 34). Hence, 200 patients (response rate = 67%) completed the study questionnaires. There were no missing data in this study, so data for 200 participants were analyzed. Table 1 presents patients’ demographical and clinical characteristics. The median patients’ age was 49.8 years (interquartile range [IQR] = 39.8–61.9). Of them, 115 (57.5%) were female, 139 (69.5%) were married, and 115 (57.5%) had a low education level (high school and less than high school). In addition, the median KPS for participants was 50.0% (IQR = 40–60). A quarter of patients (25.0%) had gastrointestinal cancer; the majority, 85.5%, received chemotherapy.

### 3.2. Decision-Making Preferences and Practices

#### 3.2.1. Patients’ Preferences

The results showed that 81 (40.5%) of the participants opted for the passive decision-making approach, 70 (35%) preferred shared decision-making (i.e., making shared decisions after consulting a physician or family), and 49 (24.5%) preferred active decision-making (i.e., to decide by himself or herself), as shown in Table 2.

The findings showed, among the participants with passive decision-making preferences, that 82 (41.0%) of participants preferred to leave the decision to their treating physician and while only 15 (7.5%) of participants preferred that the decision be taken by their family (Table 2).

#### 3.2.2. Actual Decision-Making

Concerning actual decision-making practices, the results indicated that 50% of the participants were classified as passive, 24.5% as active, and 25.5% as shared decision-makers (Figure 1). The agreement level between decision-making preferences and actual practices was significant (ⱪ coefficient = 0.69; 95% CI = 0.59 to 0.79).

#### 3.2.3. Patient Satisfaction with the Decision-Making Process

The results showed that 168 (84%) participants were satisfied with the way decisions were made, 164 (82%) patients were satisfied with the actual decisions made, and 143 (71.5%) were satisfied with the shared information, as shown in Table 3.

#### 3.2.4. Differences in Decision-Making Preferences

The results show that there were statistically significant differences in participants’ decision-control preferences according to their (1) educational level, where participants with lower educational levels had a statistically significant preference for passive and shared decision control (*p* = 0.026); (2) gender, where female participants significantly preferred passive decision-making (*p* = 0.003); and (3) religion, where Muslims significantly showed preference for passive decision-control preference (*p* = 0.011). Meanwhile, the remaining variables (age, employment, marital status, cancer type, and KPS) showed no statistically significant association with decision-control preference, as shown in Table 4.

The univariate logistic regression analysis showed that, being a male (*p* = 0.003), highly educated (college and advanced degree) (*p* = 0.018), and a Christian (*p* = 0.006) were statistically significant correlated with active decision-control preferences. Meanwhile, in multivariate logistic regression analysis, being a male (*p* = 0.008) and being a Christian (*p* = 0.017) were the only statistically significant correlations for active decision-control preferences (Table 5).

## 4. Discussion

This study demonstrated that 40.5% of patients with advanced cancer preferred passive decision control compared with 35% who preferred shared decision-making and 24.5% who preferred active decision-making. These results are consistent with the findings of published studies in different contexts [2,9]. For example, a study in Jordan reported that 89 (57%) of participating women with breast cancer preferred a passive decision role in their treatment. Of those, 57% preferred that their physicians made these decisions [2]. Similarly, the majority of enrolled Greek patients with breast cancer (71.1%) preferred a passive role in their treatment decision-making, whereas most of the recruited patients wanted to delegate the responsibility of their treatment decision to their doctor (45.3%) [22]. Meanwhile, in the study by Erin et al., 66 patients with advanced cancer mainly preferred active decision-making (39%), followed by 60 who preferred passive (35%), and 47 who preferred collaborative (27%) [23].

Some patients tend to delegate decision-making tasks to others, especially their doctors. This might be because these patients trust their primary healthcare providers. It was indicated that patients with greater trust and confidence in their physicians’ knowledge and skills desired less control over decisions, showed more satisfaction, and followed physicians’ medical advice in developed and developing countries, as shown in previous studies [24]. A further qualitative research approach is needed to understand the justification for such decision preferences.

Our study found that patients tended to choose physicians more than families to participate in their treatment decision-making process. This may reflect the complex characteristics of Jordanian families. In some cases, family members are a part of the decision-making process, and in other cases, patients may need to make the decisions alone. This may depend on the family’s well-being, culture, educational level, and composition [7,25,26,27]. In the future, further research studies are needed to address Jordanian palliative patient perception toward their physicians and families, and how it may affect their decision-making preferences.

The results showed a significant agreement between actual decision-making and decision-control preferences, which corresponded to a previously published cross-sectional survey study that showed a statistically significant correlation between actual and preferred decision-making [9]. This agreement may reflect that physicians discussed disease issues with patients appropriately, which corresponded with their preferences. However, further qualitative studies are needed to gain more insight into these results. The decision preferences varied by patient characteristics; the high frequency of passive decision control was found to be associated with a lower educational status. Furthermore, a high educational level was found to be a statistically significant correlation of active decision-control preference in univariate logistic regression analysis, but not in the multivariate logistic regression analysis. Our study findings corresponded with a previous study conducted by Colombe and colleagues [11], who found that French patients with a higher education preferred active or shared decision-making. This could be related to the fact that highly educated people are more likely to access medical information and the internet for any needed information [11]. Furthermore, a highly educated patient may better interpret information and weigh it against their preferences [28]. Moreover, mixed method studies are needed to further explore the effect of educational level and health literacy on patients’ decision-making preferences.

We also found that gender affected patient decisions significantly. Males were more willing to make an active decision than females, who preferred shared or passive decision options. This may be due to the influence of socio-cultural values, where males seem to be more empowered than females, according to the Jordanian society culture and norms. For example, in Jordan, the key person who usually makes the fateful decisions in the family is mainly a man (father, brother, or husband) [29]. However, this result needs more studies with different variables involving cultural and social aspects that may affect decision-making in relation to gender [30,31].

Most Muslim participants were reported to prefer passive decision-making. However, Jordan is a majority Muslim country; only 14 patients were Christian in this study. Additionally, we found that being a Christian was also a statistically significant correlation of the active decision-control preference. Nevertheless, the comparison between Muslims and non-Muslims is challenging, suggesting the need for further larger studies with diverse religious background stratification (e.g., Muslims vs. non-Muslims) to explore the effect of religion on patients’ decision-making preference.

Some studies have assessed the role of spirituality and religion on patients’ satisfaction. In our study, patients reported high levels of satisfaction with the decisions and the way the decisions were made regarding their cancer care. These results were similar to a pervious international cohort study, where overall satisfaction with care was related to the fact that physicians discussed care decisions in a way that was culturally appropriate for the patients and could be related to the patients’ religious background [11,32]. Hence, more qualitative studies are required to explore the factors that may affect the satisfaction of this group of patients.

Although our study revealed that most participants were satisfied with how decisions about their care were made and with the actual decisions making process, 14.5% and 14% of participants reported that they were not satisfied or undecided, respectively, regarding shared information.

It was found that advanced cancer patients preferred to receive information about their condition and detailed prognostic information. This varied not only between individuals, but also for a given individual over time. Barriers to the delivery and the understanding of information exist on both sides, for physicians and patients, in addition to family dynamics, which play an essential role as well [33]. The need for a long interaction time seems necessary to fulfill the patient’s wishes for information sharing. The communication process should be organized to ensure receiving an adequate amount of comprehensible information, which will eventually affect the decision-making process toward a more collaborative approach [24]. Furthermore, physicians must regularly ask the patient about the information they would like to know, who else should be given that information, the preferred person to be involved in the decision-making process, and how shared information should be presented [33]. Further studies are needed to evaluate patient−physician communication patterns; cultural differences; and psychosocial factors such as stress, anxiety, and depression, which may be high among patients with advanced cancer, as well as their effect on decision-control preferences [34].

The results of the current study need to be interpreted in light of the following limitations: First, the study data were collected from 2014 to 2015, which was several years ago. We do not anticipate that this would have changed over time. However, we are currently conducting another study with a larger sample size in order to be able to detect any changes in patients’ decision preferences. Second, this study was conducted in a single cancer care center, which limits the generalizability of the findings to similar settings only. Third, is the relatively small sample size compared with previous similar studies [9,18]. Fourth, a considerable number of screened patients were excluded due to the severity of their symptoms; lack of interest; or time, which correspond with Sriram and colleagues’ cross-sectional study, in that 29% of their eligible patients refused to participate in their study due to distressful symptoms and lack of time [9]. This high attrition rate may have led to selection bias and affected the generalizability of the results. Although the inclusion of those patients’ opinions is important, this attrition rate is expected in patients with advanced cancer [35]. Fifth, the directionality of associations between participants’ characteristics and decision-control preferences was unknown due to the use of a cross-sectional study design; therefore, longitudinal cohort studies are needed. Finally, there could still be residual confounding by unmeasured factors. Further mixed method studies, including all possible confounding factors, are required in order to have a more in-depth understanding of this issue in this particular cultural context.

## 5. Conclusions

In this study, most participants receiving palliative care preferred a passive role in decision-making. They preferred to delegate decision-making to their physician. Patient-related variables (gender, education, and religion) were found to affect their preferences toward participation in the decision-making process. Healthcare providers need to consider these variables when approaching patients. The results of our study represent an essential step toward patient-centered care and improved overall patient care and satisfaction. A further and deeper understanding of patients’ preferences can be sought using a qualitative research approach, which is highly recommended. Additionally, considering further variables such as patients’ psychosocial and spiritual factors and communication and information sharing from health care providers can give us more insight regarding patients’ decision-making preferences and other factors that may influence those preferences.

## Figures and Tables

**Figure 1 ijerph-20-05550-f001:**
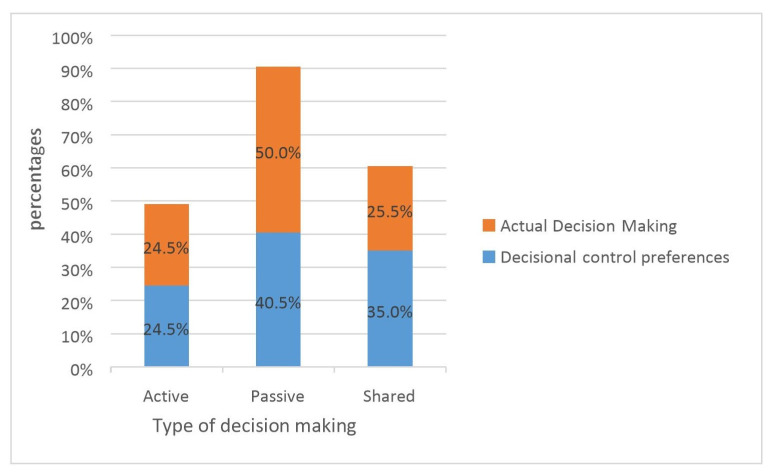
Concordance between the actual decision-making and decision-control preferences.

**Table 1 ijerph-20-05550-t001:** Patient demographics and clinical characteristics (*n* = 200).

Characteristics	*n* (%)
Gender	
Female	115 (57.5%)
Male	85 (42.5%)
Marital Status	
Divorced	5 (2.5%)
Married	139 (69.5%)
Separated	6 (3.0%)
Single	28 (14.0%)
Widowed	22 (11.0%)
Nationality	
Jordanian	200 (100%)
Age	
<65	164 (82.0%)
≥65	36 (18.0%)
median (IQR)	49.8 (39.8, 61.9)
Religion	
Muslims	186 (93%)
Christians	14 (7.0%)
Education	
College and advanced degree	85 (42.5%)
High school	69 (34.5%)
Less than high school	46 (23.0%)
Employment	
Employed	27 (13.5%)
Others	12 (6.0%)
Retired	45 (22.5%)
Unemployed	116 (58.0%)
Cancer Type	
Breast	42 (21.0%)
Gastrointestinal	50 (25.0%)
Genitourinary	13 (6.5%)
Gynecology	20 (10.0%)
Head and neck	12 (6.0%)
Hematological malignancies and others	40 (20.0%)
Lung	23 (11.5%)
Treatment modality	
Radiation	118 (59.0%)
Chemotherapy	171 (85.5%)
Immunotherapy	5 (2.5%)
Surgery	110 (55.0%)
Targeted treatment	31 (15.5%)
Karnofsky Performance status	
20	8 (4.0%)
30	17 (8.5%)
40	37 (18.5%)
50	45 (22.5%)
60	44 (22.0%)
70	27 (13.5%)
80	12 (6.0%)
90	9 (4.5%)
100	1 (0.5%)
Karnofsky Performance Status, median (IQR)	50 (40, 60)

IQR: interquartile range.

**Table 2 ijerph-20-05550-t002:** Distribution of decision role preferences according to the independent relationships between patient and physician; patient and family; and the relationships among the patient, family, and physician (*n* = 200).

Survey	*n* (%)
A. Patient and physician	
Passive	82 (41.0%)
Shared	67 (33.5%)
Active	43 (21.5%)
Do not know/prefer not to answer	8 (4.0%)
B. Patient and family	
Passive	15 (7.5%)
Shared	88 (44.0%)
Active	80 (40.0%)
Do not know/prefer not to answer	17 (8.5%)
C. Patient, family, and physician	
Passive	81 (40.5%)
Shared	70 (35.0%)
Active	49 (24.5%)

**Table 3 ijerph-20-05550-t003:** Patient satisfaction with the decision-making process (*n* = 200).

Characteristics	*n* (%)
I am satisfied with the information I receive about my care	
Completely disagree	12(6.0)
Disagree	17(8.5)
Undecided	28(14.0)
Agree	87(43.5)
Completely agree	56(28.0)
I am satisfied with the way decisions were made about my care	
Completely disagree	2(1.0)
Disagree	12(6.0)
Undecided	18(9.0)
Agree	83(41.5)
Completely agree	85(42.5)
I am satisfied with the decisions about my care	
Completely disagree	2(1.0)
Disagree	14(7.0)
Undecided	20(10.0)
Agree	84(42.0)
Completely agree	80(40.0)

**Table 4 ijerph-20-05550-t004:** Decision-control preferences based on patients’ characteristics (*n* = 200).

Name	Value	Total	Preferred Decision Control			*p*-Value (Chi-Square Test)
			Active	Passive	Shared	
Gender	Female	115 (57.5%)	19 (38.8%)	47 (58.0%)	49 (70.0%)	0.003
	Male	85 (42.5%)	30 (61.2%)	34 (42.0%)	21 (30.0%)	
Age group (years)	Age < 65	164 (82.0%)	38 (77.6%)	67 (82.7%)	59 (84.3%)	0.627
	Age ≥ 65	36 (18.0%)	11 (22.4%)	14 (17.3%)	11 (15.7%)	
Education	College & Advanced Degree	85 (42.5%)	28 (57.1%)	33 (40.7%)	24 (34.3%)	0.026
	High School	69 (34.5%)	17 (34.7%)	26 (32.1%)	26 (37.1%)	
	Less than High School	46 (23.0%)	4 (8.2%)	22 (27.2%)	20 (28.6%)	
Religion	Muslims	186 (93.0%)	41 (83.7%)	79 (97.5%)	66 (94.3%)	0.011
	Christians	14 (7.0%)	8 (16.3%)	2 (2.5%)	4 (5.7%)	
Employment	Employed	27 (13.5%)	8 (16.3%)	13 (16.0%)	6 (8.6%)	0.326
	Unemployed & Retired & Others	173 (86.5%)	41 (83.7%)	68 (84.0%)	64 (91.4%)	
Marital Status	Divorced	5 (2.5%)	0 (0%)	4 (4.9%)	1 (1.4%)	0.214
	Married	139 (69.5%)	40 (81.6%)	54 (66.7%)	45 (64.3%)	
	Separated	6 (3.0%)	2 (4.1%)	2 (2.5%)	2 (2.9%)	
	Single	28 (14.0%)	3 (6.1%)	10 (12.3%)	15 (21.4%)	
	Widowed	22 (11.0%)	4 (8.2%)	11 (13.6%)	7 (10.0%)	
Cancer Type	Breast & Gynecology	62 (31.0%)	10 (20.4%)	22 (27.2%)	30 (42.9%)	0.143
	Gastrointestinal	50 (25.0%)	14 (28.6%)	21 (25.9%)	15 (21.4%)	
	Genitourinary & Hematological Malignancies	53 (26.5%)	17 (34.7%)	20 (24.7%)	16 (22.9%)	
	Head, Neck & Lung	35 (17.5%)	8 (16.3%)	18 (22.2%)	9 (12.9%)	
Karnofsky Performance %	41–80	128 (64.0%)	33 (67.3%)	55 (67.9%)	40 (57.1%)	0.528
	≤40	62 (31.0%)	15 (30.6%)	22 (27.2%)	25 (35.7%)	
	>80	10 (5.0%)	1 (2.0%)	4 (4.9%)	5 (7.1%)	

**Table 5 ijerph-20-05550-t005:** Univariate and multivariate logistic regression using a model of active versus passive and shared decision-control preferences (*n* = 200).

Variable	Univariate OR (95% CI)	*p*-Value	Multivariate aOR (95% CI)	*p*-Value
Gender (Male vs. * Female)	2.76 (1.44–5.35)	0.003	2.54 (1.28–5.07)	0.008
Education (college and advanced degree vs. *high school and less than high school)	2.20 (1.14–4.23)	0.018	1.69 (0.85–3.37)	0.138
Religion (Muslims vs. * Christians)	0.21 (0.07–0.65)	0.006	0.24 (0.08–0.78)	0.017

OR: odd ratios; aOR: adjusted odd ratio; CI: confidence interval; *: the referrence group.

## Data Availability

Not applicable.

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
