# Peer review of "Decision-Making Preferences among Advanced Cancer Patients in a Palliative Setting in Jordan"

_ijerph, 2023, doi:10.3390/ijerph20085550_

Round 1

Reviewer 1 Report

This article is an interesting study to analyze decision-making preferences in a palliative setting in Jordan. The study demonstrate that a passive decision control preference was prominent among patients with advanced cancer in Jordan. However, I had some comments for the present manuscript as below.

C1: The authors may describe more on why is this issue important. What can health professionals benefit from knowing that a passive decision control preference was prominent among patients with advanced cancer?

C2: The authors may describe more on the reliability and the validity of the data. The data source was 299 patients. However, 99 patients refused to participate. The response rate was low and may lead to selection bias. The authors should describe more on why were these participants representative. The opinion of those who were being fatigued or lack of interest in the study was still important.

C3: The data was from 2014 to 2015, which has been several years ago. As time changes and medicine develops, are the results at that time still representative of today's trends? The authors should describe more.

C4: There have been many studies in the past on decision-making preferences. The authors should describe more similar study results and their differences from other studies.

C5: Because readers from other countries may not be familiar with the background of the Jordan health system. The authors may describe more on the feature of the Jordan health system. Information on whether medical care is easily available and how many cancer patients were diagnosed each year, should be described more. Also, what can other countries learn from the data from Jordan?

C6: The decisional preferences varied by patient characteristics, the authors mentioned that further studies are needed to evaluate decision control preference for additional variables. Specifically, what factors are not considered in this research? How to design further research.

Reviewer 2 Report

This well-written paper explores decision-making preferences among advanced cancer patients in a palliative setting in Jordan. As this is an important area of health communication research, the paper would likely interest IJERPH’s readership. In terms of English language and style, only minor editing is required for the sake of correctness, clarity and consistency, as highlighted throughout the attached PDF file.

In terms of specific feedback please note that the title is not informative enough. It is best to always name the particular population in a journal article title. I would suggest extending the title to something like this: 'Decision-making preferences among advanced cancer patients in a palliative setting in Jordan'

While the abstract is well-written, improvements could be made. Please see below for these improvements:

·       Line 18: The broader study by Yennurajalingam et al. (reference 9) is a ‘cross-sectional survey’ as opposed to a ‘prospective cross-sectional survey’. The word ‘prospective’ should be deleted.

·       Line 23: For the sake of consistency, state the percentages in brackets, not the numbers/frequencies.

·       Line 24: ‘Less educated females’ – the education and gender variables should be listed separately.

·       Line 24: ‘statistically significant association’ – Earlier in this abstract (i.e. at the end of the 'methods' part of the abstract) the particular analytical technique/statistical test used needs to be stated, along with the level of statistical significance (e.g. p<0.05).

The paper’s introduction provides clear background information, describes the relevant literature in the area, and identifies a gap in the literature. The study’s aim is related to the gap in the literature. I would suggest only minor edits to the ‘Introduction’ section:

·       Line 59: When stating ‘the majority’, specify the percentage in brackets (if this percentage is available in the referenced study).

·       Lines 67 and 68: It doesn't make sense to say ‘more than one-third’ ‘followed by’ a percentage that is also more than one third. Simply stating the percentages would be preferable.

·       Line 78: When stating ‘a study conducted in’, specify the type of study in terms of study design.

More substantial changes should be made to the ‘Materials and Methods’ section of the manuscript, most notably in relation to the ‘bivariate analysis’:

·       Line 100: The broader study by Yennurajalingam et al. (reference 9) is a ‘cross-sectional survey’ as opposed to a ‘prospective cross-sectional survey’. The word ‘prospective’ should be deleted.

·       Line 102: A recognised sampling approach (e.g. convenience sampling) should be explicitly stated.

·       Lines 184 and 185: For each of the two statistical software packages, provide the name of the software manufacturer as well as the location of the manufacturer's headquarters. This is standard practice in research papers, including in papers published in IJERPH.

·       Line 185 and 186: Specify that the listed descriptive statistics were only calculated for numeric variables.

·       Line 186: Insert a sentence stating that frequencies (i.e. numbers) and percentages were calculated for categorical variables.

·       Line 188: What type of ‘bivariate analysis’, in terms of a specific statistical test? Bivariate analysis by itself is insufficient because it does not control for potential confounding factors and does not estimate independent effects. Instead of or in addition to bivariate analysis, univariable and multivariable multinomial logistic regression analysis should be performed to adjust for potential confounding factors and estimate independent effects of participant characteristics on decision-control preferences. Multinomial logistic regression is the appropriate type of regression to use because the outcome is a nominal (rather than ordinal) categorical variable with more than two categories. If the numbers in cells of Table 4 are too small for the multinomial logistic regression, then you could consider combining categories of certain patient characteristics in a logical way (just for the regression; all categories could still be used elsewhere) and/or collapsing the three 'Preferred decisional control' categories into two categories for binary logistic regression analysis (instead of multinomial logistic regression analysis).

-          Lines 184-197: Somewhere in the 'Data Analysis' subsection, specify that a p-value<0.05 was considered statistically significant.

The ‘Results’ section of the manuscript should be updated to incorporate logistic regression results. Moreover, the ‘Results’ section should be improved upon in the following ways:

·       Line 207: Define the term 'low education level' in brackets, so as it is clear how this corresponds to the two categories 'High School' and 'Less than High School' in Table 1.

·       Line 210: In the Table 1 caption, change '(n=200)' to '(N=200)', so as to distinguish the denominator (typically expressed as N) from the numerator (typically expressed as n, as per the right-hand column of Table 1).

·       Table 1 content:

o   A space is required between each ‘n’ and ‘(%)’

o   At the bottom of the table, the median and IQR age values should be deleted because they are already stated earlier in the table.

o   An acronym definition is required below the table.

·       Line 218: 42.7% should be 41.0%.

·       Line 219: 8.2% should be 7.5%.

·       Lines 220-221: ‘When asking them to answer according to the independent relationships between patient and physician and patient and family respectively.’ – this is an incomplete sentence.

·       Line 224: In this caption, change '(n=200)' to '(N=200)', so as to distinguish the denominator (typically expressed as N) from the numerator (typically expressed as n, as per the right-hand column of Table 1).

·       Table 2 content:

o   Change 'No' to 'n'.

o   42.7% should be 41.0%.

o   34.9% should be 33.5%.

o   22.4% should be 21.5%.

o   8. 2% should be 7.5%

o   48.1% should be 44%.

o   43.7% should be 40%.

o   35 % should be 35.0% (with the decimal place specified for consistency).

·       Figure 1 content:

o   A y-axis title/label (e.g. 'Percentage') is required.

o   An x-axis title/label is required (e.g. 'Type of decision making') is required.

o   With regard to the value labels on the bars, delete the second decimal places. This is unnecessary and makes the values cut across the bar boundaries.

o   The figure, and/or its caption, need to clearly indicate the relevant denominator(s), so as it is clear what each percentage is a percentage of.

·       Line 236: '(N=200)' needs to be stated in the caption.

·       Table 3 content: Change 'N (%)' to 'n (%)'.

·       Line 244: Delete ‘gender; level of education;’ because these two variables were significant.

·       Table 4 content:

o   Change 'Age grp' to 'Age group (years)'.

o   For the cell corresponding to ‘Active and Divorced’, Insert '0 (0%)’. There shouldn't be any blank cells within this part of the table.

o   What type of statistical test produced the p-values in the far-right hand column? The type of statistical test can be indicated in the column heading or with an asterisk and footnote. As stated earlier, univariable and multivariable multinomial logistic regression should be used. The odds ratios (ORs) and corresponding 95% confidence intervals (CIs) should be reported for the univariable multinomial logistic regression models and the adjusted OR (aOR) and corresponding 95% CIs should be reported for the multivariable multinomial logistic regression model. If the numbers in any one cell of this table are too small for the multinomial logistic regression, then you could consider combining categories of certain patient characteristics in a logical way (just for the regression; all categories could still be used elsewhere) and/or collapsing the three 'Preferred decisional control' categories into two categories for binary logistic regression analysis (instead of multinomial logistic regression analysis). Please note that, if there is no logical way to combine outcome categories to give a binary variable, then multinomial logistic regression should be used instead of binary logistic regression.

o   Define the acronym 'BP1' directly below this table, or just state it in full within the table instead.

Improvements can be made in the ‘Discussion’ section of the manuscript:

·       Line 249: ‘This study demonstrated that a large portion of Jordanian patients with advanced cancer preferred passive decisional control.’ What is ‘a large portion’? Large relative to what? State the actual percentage and, ideally, compare it with a percentage from a similar past study.

·       Line 274: ‘A previously published study’ – state the type of study, in terms of study design.

·       Line 275: Delete the word ‘highly’ to change ‘highly significant’ to ‘significant’. A result in a given study cannot be 'highly significant': it either is or is not significant at the pre-defined level of significance (e.g. p<0.05). Instead of saying 'highly significant' here, it would be better to say 'statistically significant'.

·       Line 277: ‘further studies are needed’ – Given qualitative studies have been recommended earlier, state the type of studies recommended here too.

·       Lines 280-281: ‘Colombe and colleagues (2017)’ – This is an incorrect in-text reference (a number should be given in square brackets instead of the year).

·       Line 288: It is unclear what 'socio-cultural values' means. Elaborate on this and/or provide an appropriate reference.

·       Line 289: Give some examples of the relevant ‘social and cultural aspects’.

·       Lines 293-294: ‘So, the comparison between Muslims and non-Muslims is challenging, suggesting further studies to explore the effect of religion on patients’ preferred decisional making.’ - It would be best to specify 'further larger studies', given a small number in a given category (i.e. a relatively uncommon/rare event) is the particular methodological issue that needs to be overcome in future studies.

·       Line 297: Delete ‘(Table 3)’ because Tables/figures of results are not typically directly referred to in the 'Discussion' section.

·       Line 301: What types of ‘studies’?

·       Line 307: ‘A narrative literature review overwhelmingly explores’ – Who conducted this narrative literature review? Is this your own review or another review (in which case a reference would be required)? It is unclear what 'overwhelmingly explores' means. Should this just be 'found' instead?

·       Line 324: change ‘which might limit’ to 'which limits'

·       Line 326: ‘compared to previous similar studies’ – reference these studies here.

·       Line 327: ‘patients had been excluded due to the severity of their symptoms, lack of interest, or time’ – The response rate in this study should be compared with the response rates from similiar studies, with references given.

·       Lines 323-329: The following limitations also need to be highlighted:

o   The directionality of associations between participant characteristics and decision control preferences is unknown due to the use of a cross-sectional study design.

o   There is potential confounding of associations between participant characteristics and decision control preferences. Even if the recommended approach of conducting a multivariable multinomial or binary logistic regression analysis is followed, there may still be residual confounding by unmeasured factors (which would need to be stated as a limitation).

Changes should be made to the ‘Conclusion’ section of the manuscript:

·       Line 333: ‘related variables’ - specify the particular patient characteristics here.

·       Lines 323-325: The conclusion around significant patient characteristics is not supported by the results, as only bivariate associations were assessed. A type of multivariable regression, specifically multivariable multinomial or binary logistic regression, needs to be performed to adjust for potential confounding factors and estimate independent effects.

With regard to the appendices, please note that:

·       The tables in Appendices 1-5 need to be referenced within the appendices themselves, as per Appendix 6.

·       Appendix 2 needs a caption/title.

·       Appendix 5 needs a more descriptive caption/title.

Please see the attached PDF file and Word file for the changes specified above, along with minor edits to English language and style.

Round 2

Reviewer 1 Report

Thanks to the authors for answering every question in detail, I think the article has been revised and made more complete. I have no further questions.

Author Response

Dear Reviewer 1,

according to your reply, I understand that you have no further questions.

Thank you for your reply 

regards

Reviewer 2 Report

Thank you for revising this well-written, significant paper about decision-making preferences among advanced cancer patients in a palliative setting in Jordan. In the revised version of the manuscript, minor updates to wording can be made as per the attached PDF file. The most notable issue that should be amended is the use of multivariate logistic regression without using univariate logistic regression as well. Both univariate and multivariate logistic regression results should be presented, as this would allow readers to clearly see changes in effect estimates from the univariate models to the multivariate model. Therefore, with regard to the method (line 214), ‘univariate and multivariate logistic regression’ needs to be specified. Additionally, with regard to Table 5, the univariate logistic regression results need to be stated alongside the multivariate logistic regression results. Another noteworthy point in relation to Table 5 is the need to specify which category is the reference group for each variable.
